# The Effects of Pangenotypic Direct-Acting Antiviral Therapy on Lipid Profiles and Insulin Resistance in Chronic Hepatitis C Patients

**DOI:** 10.3390/v17020263

**Published:** 2025-02-14

**Authors:** Meng-Yu Ko, Yu-Chung Hsu, Hsu-Heng Yen, Siou-Ping Huang, Pei-Yuan Su

**Affiliations:** 1Division of Gastroenterology and Hepatology, Yuanlin Christian Hospital, Changhua 510012, Taiwan; kmy1978@gmail.com; 2Division of Gastroenterology and Hepatology, Changhua Christian Hospital, Changhua 500209, Taiwan; 77149@cch.org.tw (Y.-C.H.); 91646@cch.org.tw (H.-H.Y.); 182972@cch.org.tw (S.-P.H.); 3Department of Post-Baccalaureate Medicine, College of Medicine, National Chung Hsing University, Taichung 402202, Taiwan

**Keywords:** pangenotypic direct-acting antiviral, HCV, low-density lipoprotein, total cholesterol

## Abstract

Hepatitis C virus (HCV) eradication is usually associated with dyslipidemia. Most studies in this field have focused on genotype-specific direct-acting antivirals (DAAs), with research on pangenotypic DAAs being limited. This study examined how two pangenotypic DAA regimens, glecaprevir/pibrentasvir (GLE/PIB) and sofosbuvir/velpatasvir (SOF/VEL), affect lipid profiles and insulin resistance after viral eradication in chronic HCV patients. A total of 100 patients (57 with GLE/PIB and 43 with SOF/VEL) treated between September 2020 and January 2022 were included in the retrospective analysis. This study found a significant increase in LDL and TC levels after treatment (*p* < 0.001), but no significant changes in triglycerides, high-density lipoprotein, HbA1C, or the Homeostatic Model Assessment of Insulin Resistance. According to a logistic regression analysis, higher baseline LDL or TC and lower baseline glucose are predictors of the degree of increase in LDL or TC following a sustained virological response. Both pangenotypic DAA regimens significantly impact lipid profiles, particularly LDL and TC, but not insulin resistance. This study emphasizes the need for more research into the long-term metabolic effects of DAAs.

## 1. Introduction

Hepatitis C virus (HCV) infection can cause chronic hepatitis, cirrhosis, and hepatocellular carcinoma (HCC), affecting approximately 50 million people worldwide. This infection is a major public health concern, putting a heavy disease burden on both individuals and healthcare systems [1]. HCV infection is frequently asymptomatic and progresses to a chronic condition in most patients, with diagnosis usually occurring only after the disease has advanced to more severe stages of fibrosis.

The treatment of chronic HCV infection has dramatically improved with the introduction of new direct-acting antivirals (DAAs). These DAAs have higher efficacy across all genotypes and fewer side effects than previous interferon (IFN)-based therapies. While HCV eradication significantly reduces the risk of cirrhosis and HCC, patients with advanced liver fibrosis or cirrhosis must continue to undergo regular HCC screenings even after achieving a sustained virologic response (SVR) [2,3].

Several studies have found that HCV infection causes metabolic changes such as insulin resistance (IR), metabolic syndrome, and diabetes via complex pathways [4,5]. Chronic HCV infections have been linked to hepatic steatosis and dyslipidemia [6]. According to research, eradicating HCV can improve fasting glucose, HbA1C, and IR [7]. Furthermore, studies have reported varying effects on lipid profiles: some found a decrease in total cholesterol (TC) and low-density lipoprotein (LDL) levels with an increase in triglycerides (TG) during treatment, while others found an increase in LDL and TC levels post treatment but no significant changes in TG levels [8].

Most studies exploring the effect of lipid homeostasis and IR following viral eradication have focused on genotype-specific DAAs, particularly between subgroups treated with sofosbuvir (SOF)-based versus non-SOF-based DAAs [9,10]. However, limited research has investigated the impact of lipid profiles after treatment with new pangenotypic DAAs. This study aimed to evaluate the impact of viral eradication on lipid levels and IR in HCV patients after receiving pangenotypic DAA antiviral therapy.

## 2. Materials and Methods

We retrospectively included patients with chronic HCV infection who received DAA therapy between September 2020 and January 2022 at Changhua Christian Hospital in Taiwan. All patients had routine outpatient follow-ups during the treatment and none were discontinued due to adverse events. Exclusion criteria included the following: (1) patients who did not complete the lipid profile or had insufficient clinical data before and after DAA therapy; and (2) incomplete DAA therapy or a lack of sustained virological response 12 weeks after treatment. The DAA regimens consisted of glecaprevir/pibrentasvir (GLE/PIB) and sofosbuvir/velpatasvir (SOF/VEL). (Figure 1).

Lipid profiles measured TC, LDL, and TG. HCV genotyping was performed on all patients. A complete blood count was taken, liver function tests were conducted, and HCV RNA levels were measured before and 12 weeks after treatment. The current study was authorized by the institutional review board (IRB No. 231011), and informed consent was waived due to the anonymization of all data.

Liver elastography, steatosis, lipid profile, fasting glucose, and insulin levels were measured before and 12 weeks after DAA therapy. The degree of liver fibrosis was determined using FibroScan^®^ 530 compact (Echosens, Franceand) and the FIB-4 index, which was calculated from indirect serum markers. Advanced fibrosis was defined as an FIB-4 score above 3.25 [11]. Liver stiffness and steatosis were determined via transient elastography, with a measurement range of 2.5 kPa to 75 kPa, and the controlled attenuation parameter (CAP), which ranges from 100 to 400 decibels per meter (dB/m), with the FibroScan^®^ compact 530 (Echosens, Paris, France). Insulin resistance was assessed using the Homeostatic Model Assessment of Insulin Resistance (HOMA-IR), which is calculated as (fasting glucose level × fasting serum insulin level)/405 [12]. The percentage change in LDL and TC was calculated by subtracting the pretreatment value from the post-treatment value and dividing by the pretreatment value.

### Statistical Analysis

Demographic and other clinical data for continuous variables are given as mean ± standard deviation, while categorical variables are given as numbers and percentages. Baseline data comparisons between the GLE/PIB and SOF/VEL groups were made using the Chi-square test or Fisher’s exact test for categorical variables, and Student’s t-test for continuous variables. The paired samples t-test was used to compare the mean values of continuous data at two different time points: baseline (T0) and 12 weeks after the completion of DAA therapy (SVR). Pearson’s correlation coefficient was used for correlation analysis. Logistic regression models were used in both univariate and multivariate analyses. Factors that were significantly associated in univariate analyses were included in the multivariate model through backward elimination. All statistical analyses were carried out using PASW Statistics version 18 (formerly SPSS; IBM, Hong Kong). A *p*-value below 0.05 was deemed statistically significant.

## 3. Results

### 3.1. Baseline Characteristics of Total and Subgroup Patients

Our study included 100 patients: 57 treated with GLE/PIB and 43 treated with SOF/VEL. Table 1 shows the baseline characteristics in detail. Of the 100 patients, 50 (50%) were male, with an average age of 58.6 ± 12.7 years. The majority of patients (49%) had genotype 2 HCV. The average HCV RNA level was 5.53 ± 1.2 log10 IU/mL. Additionally, 36 patients (36%) had hypertension, 12 (12%) had diabetes mellitus, and 12 (12%) had cancer, including oral cancer (n = 3), colon cancer (n = 3), breast cancer (n = 2), lung cancer (n = 2), hepatoma (n = 1), and lymphoma (n = 1). Nineteen patients (19%) had an FIB-4 score ≥ 3.25. When comparing the two subgroups (GLE/PIB and SOF/VEL), the SOF/VEL group had a higher proportion of males, higher Aspartate Transaminase (AST) levels, lower LDL and TC levels, and greater liver stiffness.

### 3.2. Results Before and After DAA Treatment of the Total Patients

The laboratory tests and elastography results for the 100 patients before and after HCV treatment are shown in Table 2. Liver stiffness and FIB-4 scores significantly decreased (*p* < 0.001), whereas CAP increased after DAA therapy (*p* = 0.007). After treatment, LDL and TC levels were significantly higher than before (*p* < 0.001). There were no significant differences in fasting glucose, insulin, HOMA-IR, HbA1c, or HDL before and after treatment. The TG level was slightly higher after treatment than before, but the difference was not statistically significant (*p* = 0.088).

### 3.3. Results Before and After DAA Treatment for Subgroups of Two Pangenotypic DAAs

Table 3 shows the analysis of the two subgroups, separated by their DAA regimens: 57 patients received glecaprevir/pibrentasvir (GLE/PIB), while 43 received sofosbuvir/velpatasvir (SOF/VEL). TC and LDL levels were significantly higher after treatment in both subgroups than before treatment, with the GLE/PIB group showing a more pronounced increase. While liver stiffness measurements significantly decreased in both groups following treatment, the controlled CAP only increased in the GLE/PIB group.

### 3.4. Factors Associated with the Change in LDL and TC in Entire Population

Only pretreatment fasting glucose and LDL were found to be negatively associated with the percentage change in LDL after treatment in both univariate and multivariate analyses (*p* = 0.005 and *p* = 0.007, respectively). (Table 4) The findings were similar for TC, with pretreatment fasting glucose and TC being negatively associated with the percentage change in total cholesterol after treatment in both univariate and multivariate analyses (***p*** = 0.001 and *p* = 0.009, respectively) (Table 5). Figure 2 shows a negative correlation between baseline LDL and the percentage change in LDL (r = −0.273), as well as between baseline TC and the percentage change in TC (r = −0.301).

## 4. Discussion

This is the first study to look at the effects of two different pangenotypic DAA regimens, GLE/PIB and SOF/VEL, on lipid profiles and IR. Our results show that LDL and TC levels were increased after viral eradication by two pangenotypic DAAs, but there were no significant changes in fasting glucose, HbA1c, HOMA-IR, TG, or HDL.

HCV can increase lipid biosynthesis and lower the export of apolipoproteins via multiple mechanisms of lipid metabolism. This phenomenon is reversed after HCV is eradicated using DAAs [13]. There is some debate about which lipid profiles are affected by HCV infection. Most studies show that LDL and TC are the most commonly affected lipids following HCV eradication [6,7,8]. However, some studies show that HCV eradication by DAA therapy lowers triglycerides and raises HDL levels [14,15]. Our study found that LDL and TC levels increased after HCV eradication, which is consistent with most previous studies. Furthermore, the effects of various DAA regimens on lipid profiles have yielded inconsistent results. Inoue Takako et al. and Endo Daisuke et al. found that patients treated with sofosbuvir plus ledipasvir had higher LDL and TC levels than those treated with daclatasvir plus asunaprevir [10,16]. A review by YW Wang et al. discovered that SOF-based DAAs cause more significant increases in LDL than non-SOF-based DAAs [17]. However, our findings indicated that SOF-based (SOF/VEL) and non-SOF-based (GLE/PIB) DAAs have similar effects on lipid profiles after viral eradication. LDL and TC levels significantly increased following DAA therapy, while TG and HDL levels remained unchanged. This could be explained by the strong and comparable efficacy of both pangenotypic DAAs in eliminating HCV, resulting in similar effects on lipid profiles.

Several factors have been linked to increased levels of LDL and TC [7]. These include the absence of cirrhosis, hyperlipidemia, and a larger baseline waist circumference [18]. Additionally, higher HOMA-IR, lower AST, higher triglycerides, and a higher BMI at baseline were linked to changes in LDL levels [15,16]. In our study, the predictors of the degree of change in LDL and TC included fasting glucose and baseline LDL and TC levels. These findings are consistent with those of previous research, but more studies are needed to confirm them.

Previous research has shown that IR is closely associated with chronic hepatitis C infection, particularly in genotypes 1 and 4 [5]. DAA therapy can reverse IR in patients with chronic hepatitis C infection and improve hyperglycemia [19]. One mechanism linking HCV and IR is that the HCV core protein induces serine phosphorylation of the insulin receptor substrate protein. This process inhibits phosphatidylinositol-4,5-bisphosphate 3-kinase signaling, which is followed by a decrease in protein kinase B and tuberous sclerosis complex 1/2 signaling [20,21]. Our study found no significant difference in fasting glucose, HbA1C, or HOMA-IR levels between baseline and post-treatment. However, there was a trend of mild decreases in HbA1C and HOMA-IR following DAA treatment. One possible explanation is that our study’s baseline levels of fasting glucose, HbA1C, and HOMA-IR were lower than those in other studies, resulting in no significant differences [19,22].

This study has several limitations. First, the sample size was small because we only included patients who had complete data on fasting glucose, insulin, HbA1C, and lipid profiles before and after DAA treatment. Because this was a real-world retrospective study, less than half of the patients who were initially included were analyzed. Consequently, some older patients or those with multiple comorbidities were excluded, which could explain why some parameters showed no significant differences. Second, the follow-up period in our study was brief. Researchers discovered that LDL and TC levels remained elevated even two years after DAA therapy. However, most studies observed a reduction in carotid atherosclerosis after HCV eradication [7]. In addition, previous studies have shown that HCV infection can increase the risk of cardiovascular disease, while antiviral therapy can lower this risk [23,24]. The possible mechanism is multifactorial, involving lipid disturbances, vascular injury, oxidative stress, and endothelial dysfunction. Therefore, more research is needed to understand the long-term impact of pangenotypic DAAs and their influence on clinical outcomes.

## 5. Conclusions

Treatment with the two pangenotypic DAAs increased LDL and TC levels. Our study found no significant differences in glucose, HbA1c, HOMA-IR, or TG levels before or after DAA therapy. The only significant predictors of an increase in LDL or TC levels at SVR12 were lower baseline fasting glucose and lower LDL or TC levels.

## Figures and Tables

**Figure 1 viruses-17-00263-f001:**
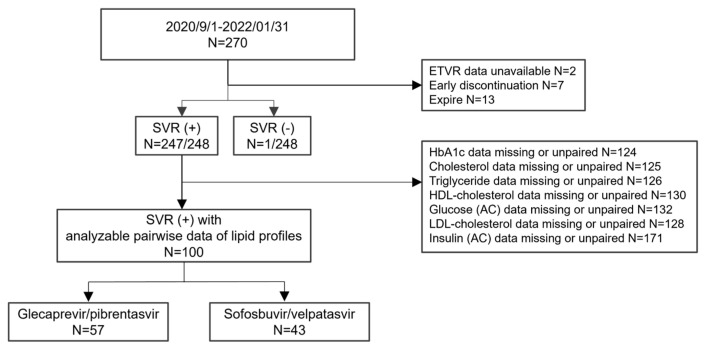
The flowchart of participants included and excluded from the study.

**Figure 2 viruses-17-00263-f002:**
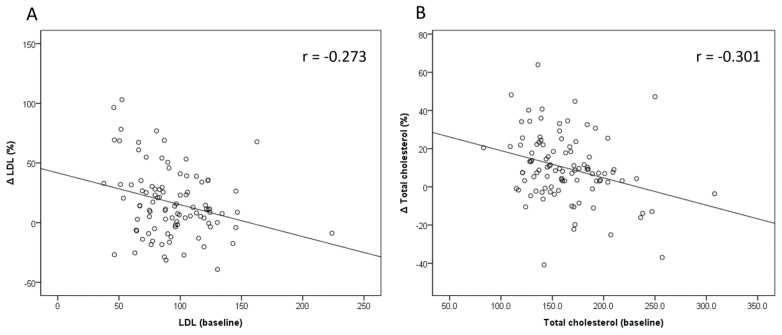
The correlation between pretreatment LDL and TC levels and the percentage change of (**A**) LDL (ΔLDL(%)) and (**B**) TC (ΔTC(%)).

**Table 1 viruses-17-00263-t001:** The baseline characteristics of the patients in the study group.

Variable	Total (N = 100)	GLE/PIB (N = 57)	SOF/VEL (N = 43)	*p*-Value
Age, yrs	58.6 ± 12.7	57.1 ± 11.3	60.6 ± 14.2	0.173
Gender (Male), n (%)	50 (50.0%)	22 (38.6%)	28 (65.1%)	0.009
Height, cm	160.6 ± 9	159.9 ± 9.9	161.4 ± 7.7	0.429
Body weight, kg	64.6 ± 12.4	65.7 ± 14.1	63.2 ± 9.7	0.306
BMI, kg/m^2^	24.96 ± 3.65	25.47 ± 3.7	24.29 ± 3.52	0.111
DM, n (%)	12 (12.0%)	6 (10.5%)	6 (14.0%)	0.602
Hypertension, n (%)	36 (36.0%)	18 (31.6%)	18 (41.9%)	0.289
HCV RNA, log10 IU/mL	5.53 ± 1.2	5.74 ± 1.06	5.25 ± 1.34	0.055
Genotype, n (%)				0.566
1/1a/1b	37 (37.0%)	24 (42.1%)	13 (30.2%)	
2	49 (49.0%)	24 (42.1%)	25 (58.1%)	
3	4 (4.0%)	2 (3.5%)	2 (4.7%)	
6	4 (4.0%)	3 (5.3%)	1 (2.3%)	
Indeterminate	6 (6.0%)	4 (7.0%)	2 (4.7%)	
GOT (AST), U/L	69.2 ± 94.4	47.1 ± 27.8	98.5 ± 135.7	0.018
GPT (ALT), U/L	95.8 ± 198.5	64.1 ± 44.8	137.9 ± 295	0.111
Platelet count, 10^9^/L	203.8 ± 63.6	208.6 ± 62.8	197.4 ± 64.7	0.386
Hb, g/dl	13.9 ± 1.7	14 ± 1.5	13.8 ± 1.8	0.514
INR	1 ± 0.15	0.99 ± 0.19	1 ± 0.07	0.664
Bilirubin-T, mg/dl	0.72 ± 0.36	0.69 ± 0.21	0.77 ± 0.49	0.309
Creatinine, mg/dl	0.99 ± 0.94	0.91 ± 0.84	1.09 ± 1.06	0.345
Albumin, g/dl	4.2 ± 0.3	4.2 ± 0.3	4.1 ± 0.3	0.146
FIB-4	2.78 ± 3.73	2.43 ± 4.17	3.23 ± 3.03	0.293
Glucose (AC), mg/dL	103.5 ± 21.6	104.3 ± 19.6	102.4 ± 24.1	0.665
Insulin (AC), μIU/mL	9.79 ± 8.95	8.9 ± 7.84	11.57 ± 10.78	0.215
HOMA-IR	2.69 ± 3.2	2.3 ± 2.12	3.47 ± 4.62	0.228
HbA1c, %	5.8 ± 0.8	5.7 ± 0.9	5.8 ± 0.8	0.65
HDL-cholesterol, mg/dL	47.2 ± 15.5	48.5 ± 16.4	45.4 ± 14.3	0.318
LDL-cholesterol, mg/dL	93.4 ± 29.3	98.8 ± 32	86.7 ± 24.3	0.044
Total cholesterol, mg/dL	163 ± 36.9	171.3 ± 38.2	152 ± 32.5	0.009
Triglyceride, mg/dL	118.7 ± 81.2	126.4 ± 99.6	108.5 ± 46.2	0.234
Baseline Fibrosis (FIB-4), n (%)				0.003
F0-2	81 (81.0%)	52 (91.2%)	29 (67.4%)	
F3-4 (FIB-4 ≥ 3.25)	19 (19.0%)	5 (8.8%)	14 (32.6%)	
LSM using FibroScan, kPa	8.8 ± 7	7.2 ± 3	11.5 ± 10.4	0.037
CAP using FibroScan, dB/m	228 ± 43	233 ± 46	221 ± 38	0.225

**Table 2 viruses-17-00263-t002:** The results of 100 patients before and after HCV treatment with direct-acting antivirals. (Data are expressed as mean ± standard deviation.).

Variable	Time Point	Value	*p*-Value
LSM using FibroScan, kPa	T0	8.699 ± 6.289	<0.001
	SVR	6.561 ± 4.026	
CAP using FibroScan, dB/m	T0	227.522 ± 39.126	0.007
	SVR	241.657 ± 50.303	
FIB-4	T0	2.776 ± 3.729	0.021
	SVR	2.024 ± 1.424	
Fasting glucose, mg/dL	T0	103.5 ± 21.568	0.251
	SVR	109.65 ± 49.937	
Fasting insulin, μIU/mL	T0	10.06 ± 9.07	0.719
	SVR	9.66 ± 6.43	
HOMA-IR	T0	2.77 ± 3.26	0.571
	SVR	2.56 ± 1.83	
HbA1c, %	T0	5.779 ± 0.841	0.162
	SVR	5.711 ± 0.761	
HDL-cholesterol, mg/dL	T0	47.19 ± 15.539	0.440
	SVR	48 ± 13.924	
LDL-cholesterol, mg/dL	T0	93.41 ± 29.343	<0.001
	SVR	106.777 ± 36.472	
Total cholesterol, mg/dL	T0	163.02 ± 36.93	<0.001
	SVR	177.33 ± 41.738	
Triglyceride, mg/dL	T0	118.7 ± 81.24	0.088
	SVR	129.83 ± 98.46	
Body weight, kg	T0	65.1 ± 13.4	0.288
	SVR	64.7 ± 13.3	
BMI, kg/m^2^	T0	25.07 ± 3.57	0.192
	SVR	24.88 ± 3.63	

**Table 3 viruses-17-00263-t003:** Results before and after DAA treatment for subgroups of two pangenotypic DAAs (GLE/PIB and SOF/VEL).

GLE/PIB (N = 57)				SOF/VEL (N = 43)			
Variable	Time Point	Value	*p*-Value	Variable	Time Point	Value	*p*-Value
LSM using FibroScan, kPa	T0	7.467 ± 3.141	<0.001	LSM using FibroScan, kPa	T0	10.904 ± 9.358	0.001
	SVR	5.802 ± 2.058			SVR	7.921 ± 5.981	
CAP using FibroScan, dB/m	T0	230.186 ± 40.9	0.014	CAP using FibroScan, dB/m	T0	222.75 ± 36.071	0.241
	SVR	246.605 ± 53.344			SVR	232.792 ± 44.014	
FIB-4	T0	2.434 ± 4.174	0.215	FIB-4	T0	3.23 ± 3.029	0.001
	SVR	1.77 ± 1.032			SVR	2.361 ± 1.776	
Glucose (AC), mg/dL	T0	104.316 ± 19.624	0.953	Glucose (AC), mg/dL	T0	102.419 ± 24.102	0.250
	SVR	104.421 ± 15.004			SVR	116.581 ± 74.106	
Insulin (AC), μIU/mL	T0	9.08 ± 7.99	0.793	Insulin (AC)	T0	11.97 ± 10.81	0.437
	SVR	9.4 ± 6.36			SVR	10.18 ± 6.67	
HOMA-IR	T0	2.35 ± 2.17	0.717	HOMA-IR	T0	3.6 ± 4.67	0.353
	SVR	2.47 ± 1.8			SVR	2.73 ± 1.9	
HbA1c, %	T0	5.746 ± 0.863	0.199	HbA1c, %	T0	5.823 ± 0.818	0.46
	SVR	5.675 ± 0.613			SVR	5.758 ± 0.927	
HDL-cholesterol, mg/dL	T0	48.544 ± 16.384	0.487	HDL-cholesterol, mg/dL	T0	45.395 ± 14.333	0.715
	SVR	49.526 ± 12.696			SVR	45.977 ± 15.321	
LDL-cholesterol, mg/dL	T0	98.8 ± 32	<0.001	LDL-cholesterol, mg/dL	T0	86.698 ± 24.34	0.042
	SVR	116.5 ± 38			SVR	94.53 ± 30.768	
Total cholesterol, mg/dL	T0	171.3 ± 38.2	<0.001	Total cholesterol, mg/dL	T0	152.047 ± 32.467	0.025
	SVR	188.9 ± 42.8			SVR	161.977 ± 35.242	
Triglyceride, mg/dL	T0	126.404 ± 99.63	0.306	Triglyceride, mg/dL	T0	108.488 ± 46.203	0.109
	SVR	136.579 ± 113.994			SVR	120.884 ± 73.355	
Body weight, kg	T0	66.482 ± 14.773	0.578	Body weight, kg	T0	62.5 ± 10.4	0.361
	SVR	66.271 ± 14.879			SVR	61.8 ± 9.6	
BMI, kg/m^2^	T0	25.485 ± 3.762	0.773	BMI, kg/m^2^	T0	24.33 ± 3.12	0.098
	SVR	25.436 ± 3.801			SVR	23.89 ± 3.13	

**Table 4 viruses-17-00263-t004:** Univariate and multivariate analyses of factors associated with the percentage of change in LDL-cholesterol.

Variables	Univariate				Multivariate			
	B	95% CI for B		*p*-Value	B	95% CI for B		*p*-Value
		Lower	Upper			Lower	Upper	
Age	−0.004	−0.009	0.000	0.068				
Gender	−0.044	−0.160	0.071	0.449				
BMI	−0.008	−0.025	0.008	0.335				
DM	−0.036	−0.212	0.140	0.687				
HT	−0.088	−0.207	0.031	0.145				
CANCER	−0.057	−0.240	0.126	0.536				
HCV RNA (log10)	0.021	−0.027	0.069	0.396				
Genotype								
Non-GT1	--	--	--	--				
GT1	−0.004	−0.125	0.116	0.942				
GOT (AST)	0.0001	−0.001	0.001	0.797				
GPT (ALT)	0.0001	0.000	0.000	0.575				
Platelet count	−0.001	−0.001	0.000	0.251				
Hb	−0.004	−0.038	0.031	0.842				
I.N.R.	−0.218	−0.605	0.169	0.267				
Bilirubin-T	−0.168	−0.325	−0.010	0.037				
Creatinine	0.026	−0.035	0.087	0.397				
Albumin	−0.088	−0.256	0.081	0.305				
Glucose (AC)	−0.004	−0.007	−0.002	0.002	−0.004	−0.006	−0.001	0.005
Insulin (AC)	−0.002	−0.009	0.006	0.635				
HbA1c, %	−0.024	−0.093	0.044	0.480				
Cholesterol	−0.001	−0.003	0.000	0.171				
LDL-cholesterol	−0.003	−0.005	−0.001	0.007	−0.003	−0.004	−0.001	0.007
HDL-cholesterol	0.003	−0.001	0.006	0.150				
Triglyceride	0.00004	−0.001	0.001	0.942				
DAA regimen								
GLE/PIB	--	--	--	--				
SOF/VEL	−0.093	−0.208	0.023	0.113				
LSM	0.0001	−0.009	0.009	0.985				
CAP	−0.001	−0.003	0.001	0.194				
FIB-4	0.003	−0.012	0.019	0.677				

**Table 5 viruses-17-00263-t005:** Univariate and multivariate analyses of factors associated with the percentage change in total cholesterol.

Variables	Univariate				Multivariate			
	B	95% CI for B		*p*-Value	B	95% CI for B		*p*-Value
		Lower	Upper			Lower	Upper	
Age	−0.0004	−0.003	0.002	0.784				
Gender	−0.014	−0.084	0.056	0.686				
BMI	−0.005	−0.014	0.005	0.352				
DM	−0.009	−0.117	0.098	0.865				
HT	−0.025	−0.098	0.047	0.487				
CANCER	−0.026	−0.133	0.082	0.637				
HCV RNA(log10)	0.009	−0.020	0.038	0.537				
Genotype								
Non-GT1	--	--	--	--				
GT1	0.036	−0.036	0.108	0.327				
GOT (AST)	−0.00001	0.000	0.000	0.974				
GPT (ALT)	0.00002	0.000	0.000	0.857				
Platelet count	−0.00027	−0.001	0.000	0.323				
Hb	−0.001	−0.022	0.020	0.937				
I.N.R.	−0.327	−0.555	−0.099	0.005				
Bilirubin-T	−0.070	−0.167	0.028	0.159				
Creatinine	0.016	−0.022	0.053	0.409				
Albumin	−0.026	−0.129	0.077	0.614				
Glucose (AC)	−0.003	−0.004	−0.001	<0.001	−0.002	−0.004	−0.001	0.001
Insulin (AC)	−0.001	−0.005	0.003	0.487				
HbA1c, %	−0.026	−0.067	0.016	0.222				
Cholesterol	−0.001	−0.002	−0.001	0.002	−0.001	−0.002	0.000	0.009
LDL-cholesterol	−0.001	−0.002	0.000	0.126				
HDL-cholesterol	−0.002	−0.004	0.001	0.138				
Triglyceride	−0.0004	−0.001	0.000	0.039				
DAA regimen								
GLE/PIB	--	--	--	--				
SOF/VEL	−0.034	−0.104	0.036	0.339				
LSM	0.001	−0.004	0.007	0.584				
CAP	−0.001	−0.001	0.000	0.197				
FIB-4	0.003	−0.006	0.012	0.532				

## Data Availability

The data that support the findings of this study are available from the corresponding author (P.S) upon reasonable request.

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
