# Peer review of "The Effects of Pangenotypic Direct-Acting Antiviral Therapy on Lipid Profiles and Insulin Resistance in Chronic Hepatitis C Patients"

_viruses, 2025, doi:10.3390/v17020263_

Round 1
Reviewer 1 Report
Comments and Suggestions for Authors
The major point of concern is how the effect of HCV eradication and the effect of a rather short course of DAAs on cholesterol is discussed. How can the authors be certain, that DAA treatment and not HCV eradication is the cause of the increase in LDL- and total cholesterol at SVR. A pharmacological effect attributed to the drug should be limited to treatment duration. A historical cohort of patients with SVR after interferon based therapy for comparison would be helpful to add some information on this topic. However, given the fact that interferon based therapies typically induces weight loss and is selective for patients responding with SVR compared to DAAs this approach may have some limitations. Numerous studies have shown an effect of HCV eradication on lipid levels limiting the novelty of this paper.
Minor comments:
Specify cancer. Define resistance and CAP thresholds in the method section.
Author Response
Comments 1: The major point of concern is how the effect of HCV eradication and the effect of a rather short course of DAAs on cholesterol is discussed. How can the authors be certain, that DAA treatment and not HCV eradication is the cause of the increase in LDL- and total cholesterol at SVR. A pharmacological effect attributed to the drug should be limited to treatment duration. A historical cohort of patients with SVR after interferon-based therapy for comparison would be helpful to add some information on this topic. However, given the fact that interferon based therapies typically induces weight loss and is selective for patients responding with SVR compared to DAAs this approach may have some limitations. Numerous studies have shown an effect of HCV eradication on lipid levels limiting the novelty of this paper.
Response 1:
Thank you for pointing this out regarding the effects of HCV eradication versus the pharmacological effects of DAAs on lipid profiles. We believe the observed changes are primarily due to HCV eradication because previous studies have shown that chronic HCV infection is associated with metabolic changes, such as dyslipidemia and decreased lipid biosynthesis. After HCV eradication, lipid metabolism tends to return to pre-infection levels. Some studies have also shown that lipid changes persist post-treatment, supporting the hypothesis that DAAs primarily act to eliminate HCV, while pharmacological effects would diminish upon discontinuation of therapy. Previous studies on interferon-based therapy have also demonstrated changes in lipid profiles after HCV eradication, though the results remain debated. Further research is needed to directly compare long-term lipid profile changes between DAA and interferon-based therapy to clarify these effects.
Because most of the previous studies focused on genotype-specific direct-acting antivirals, the novelty of this study lies in comparing two pangenotypic DAAs.
Comments 2: Specify cancer. Define resistance and CAP thresholds in the method section.
Response 2:
Thank you for pointing this out. The changes can be found on page 2, lines 73-75, and on page 3, lines 101-102. “Additionally, 36 patients (36%) had hypertension, 12 (12%) had diabetes mellitus, and 12 (12%) had cancer, including oral cancer (n=3), colon cancer (n=3), breast cancer (n=2), lung cancer (n=2), hepatoma (n=1), and lymphoma (n=1).” And “Liver stiffness and steatosis were determined via transient elastography, with a measurement range of 2.5 kPa to 75 kPa, and the controlled attenuation parameter (CAP), which ranges from 100 to 400 decibels per meter (dB/m), using the FibroScan® compact 530 (Echosens, France).”
Reviewer 2 Report
Comments and Suggestions for Authors
Meng-Yu Ko et al. investigated the effect of glecaprevir/pibrentasvir and sofosbuvir/velpatasvir on lipid profiles and insulin resistance in chronic HCV patients. This study found that the low‑density lipoprotein (LDL) cholesterol and total cholesterol (TC) levels were increased in patients after treatment with these drugs, suggesting that the antiviral treatment may dysregulate metabolic profile of HCV patients. The study is interesting. However, the conclusion might not be fully supported. The specific comments are listed below.
1. In addition to HCV, other hepatotropic viruses can also establish chronic infection in the human liver. It would be helpful to clarify whether these patients were positive for HBV or HEV. If the patients were positive for more than two hepatotropic viruses, I believed that the conclusion of this study might not be sound. Please provide more information regarding the co-infection of these patients with other hepatotropic viruses.
2. In the abstract, “While Hepatitis C virus (HCV) eradication usually results in higher levels of low‑density lipoprotein (LDL) cholesterol and total cholesterol (TC), research on the effects of pangenotypic direct‑acting antivirals (DAAs) is limited.” This sentence is confusing. Please modify.
3. In the abstract, the authors found an increase in lipid profile. However, the authors stated that “Both pangenotypic DAA regimens have a negative impact on lipid profiles”. The statement is not clear. Please modify.
Comments on the Quality of English LanguageThe manuscript is fine, but the abstract needs to be polished.
Author Response
Comments 1: In addition to HCV, other hepatotropic viruses can also establish chronic infection in the human liver. It would be helpful to clarify whether these patients were positive for HBV or HEV. If the patients were positive for more than two hepatotropic viruses, I believed that the conclusion of this study might not be sound. Please provide more information regarding the co-infection of these patients with other hepatotropic viruses.
Response 1: Thank you for pointing this out. Only four patients had HBV-HCV coinfection. We have not conducted HEV testing at our institution, but the prevalence of HEV infection is low in our country.
Comments 2: In the abstract, “While Hepatitis C virus (HCV) eradication usually results in higher levels of low‑density lipoprotein (LDL) cholesterol and total cholesterol (TC), research on the effects of pangenotypic direct‑acting antivirals (DAAs) is limited.” This sentence is confusing. Please modify.
Response 2: Thanks for your suggestions. We have modified the sentences as follows: Hepatitis C virus (HCV) eradication is commonly associated with dyslipidemia. Most studies have focused on genotype-specific direct-acting antivirals (DAAs), while research on pangenotypic DAAs remains limited. The changes can be found on page 1, lines 13-15.
Comments 3: In the abstract, the authors found an increase in lipid profile. However, the authors stated that “Both pangenotypic DAA regimens have a negative impact on lipid profiles”. The statement is not clear. Please modify.
Response 3: Thanks for your kind recommendation. We have modified the sentences as follows: Both pangenotypic DAA regimens have a significant impact on lipid profiles, particularly on LDL and TC, but not on insulin resistance. The changes can be found on page 1, lines 24-25.
Round 2
Reviewer 1 Report
Comments and Suggestions for Authors
The wording in the introduction still implies that the change in lipids is due to the DAA regimen and not HCV eradication as stated in the authors response. This should be definitely changed to avoid misinterpretations.
An important aspect missing in the discussion is that HCV infection may result in a higher incidence of cardiovascular events, e.g. Butt AA et al. CID 2017;65(4):557–565. This may be due to other biological effects beyond simple effects on lipids. The authors should add this more complex view to their discussion.
Some minor edits in English would help.
Author Response
Comments 1: The wording in the introduction still implies that the change in lipids is due to the DAA regimen and not HCV eradication as stated in the authors response. This should be definitely changed to avoid misinterpretations.
Response 1:
Thank you for pointing this out. We have made another change to the introduction on page 2, lines 52–57. “Most studies exploring the effect of lipid homeostasis and IR following viral erad-ication have focused on genotype-specific DAAs, particularly between subgroups treated with sofosbuvir (SOF)-based versus non-SOF-based DAAs [9,10]. However, limited research has investigated the impact of lipid profiles after treatment with new pangenotypic DAAs. This study aimed to evaluate the impact of viral eradication on lipid levels and IR in HCV patients after receiving pangenotypic DAA antiviral therapy.”
Comments 2: An important aspect missing in the discussion is that HCV infection may result in a higher incidence of cardiovascular events, e.g. Butt AA et al. CID 2017;65(4):557–565. This may be due to other biological effects beyond simple effects on lipids. The authors should add this more complex view to their discussion.
Response 2: Thank you for your suggestion. We have added a discussion about the effect of HCV infection on cardiovascular disease and cited the references as you suggested. Changes have been made on page 12-13, lines 220-226. “Researchers discovered that LDL and TC levels remained elevated even two years after DAA therapy. However, most studies observed a reduction in carotid atherosclerosis after HCV eradication [7]. In addition, previous studies have shown that HCV infection can increase the risk of cardiovascular disease, while antiviral therapy can lower this risk [23,24]. The possible mechanism is multifactorial, involving lipid disturbances, vascular injury, oxidative stress, and endothelial dysfunction.”
Reviewer 2 Report
Comments and Suggestions for Authors
The response letter addressed my concerns, and I believe the manuscript is now ready for publication.
Author Response
Comments 1: The response letter addressed my concerns, and I believe the manuscript is now ready for publication.
Response 1:
Thank you for your suggestion.